

# Incorporating science communication and bicultural knowledge in teaching a blended volcanology course

Ben Kennedy[1], Kamen Engel[1], Jonathan Davidson[1], Sylvia Tapuke[2], Dan Hikuroa[2], Tim Martin[3], Pinelopi Zaka[4]

[1] School of Earth and Environment, University of Canterbury, Christchurch, 8041, Aotearoa New Zealand
[2] Te Wānanga ō Waipapa—Māori Studies, University of Auckland, Auckland 1010, Aotearoa New Zealand
[3] Elon University, Elon, North Carolina USA
[4] AsureQuality, Christchurch, Aotearoa New Zealand

Correspondence to:

 <ben.kennedy@canterbury.ac.nz>

**Abstract.** A variety of skills can be taught alongside course content. In the University of Canterbury third-year
course on magmatic systems and volcanology, we chose to focus on teaching bicultural competence and science
communication while transforming the course to a more skills-based, flexible, flipped classroom model. We
document the development process and measure student perceptions associated with these skills. We used two
edX massive open online volcanology courses (MOOCs) as skills-focussed learning resources to replace lectures
and supplement hands-on laboratory and tutorial sessions to teach volcanology.
We compare the skills-focussed courses with baseline data from 2021, gathered during an initial iteration of the
course which included interactive volcanology lectures, and an online Iceland virtual fieldtrip component. The
new course was developed using the original 2021 Iceland virtual fieldtrip to create the two virtual fieldtrip-based
MOOCs with new bicultural and science communication components. To achieve this, we used cultural advisors
from connections through Aotearoa New Zealand research programs and kaiārahi (Māori learning advisors) from
the University of Canterbury. In the course, these experts ensured appropriate cultural guidance at specific
volcanic sites and appropriate assessments. Mātauranga (Māori knowledge) of volcanoes is included and taught
by video of kōrero (oral knowledge) from members of mana whenua (tribes local to the volcanoes) in the areas
that are visited in the course.
In this paper we describe the development of a flipped classroom MOOC featuring bicultural competence and
science communication skills, and we report students' reflections on learning with a focus on these featured points.
We analyse student reflections and comments from the two iterations of the online content by specifically coding
for comments regarding skills learnt. Student responses to the reflective question *What did you learn in this course*
*and why is it important to you and/or your potential career?* show a marked shift between the years. In the new
2022 course, students' reflections were more likely to highlight a skill rather than content, and there was a large
increase in students who reported science communication or bicultural competence as a potential skill that would
be useful to them. Student quotes from throughout the course and in response to the reflective question- *Has this*
*course influenced your bicultural competence?* are used to explore how and why these skills were valued by the
students. These courses provide a freely available and potentially flexible model to teach bicultural and science
communication skills alongside volcanology.
**1 Introduction.**
Geoscience instructors teach content and skills in a variety of settings, e.g., lectures, laboratories, online modules,
projects, and in person and virtual field experiences. To learn and master new skills students need to employ
adaptive expertise techniques (Bohle Carbonell et al., 2016) and distributed practice (Benjamin & Tullis, 2010)
by practicing skills multiple times and in different scenarios. Bicultural confidence is a valued skill in Aotearoa
New Zealand[1], and cultural confidence and competence is important globally and inherently linked to sense of
place, a key concept in the geosciences (Apple et al., 2014). However, cultural competence is rarely taught in
geoscience courses (Mosher and Keane, 2021). Here, we present how we integrated a new skills-based learning
goal (including bicultural confidence) into the course. This additional learning goal was introduced and assessed
in a 3rd year volcanology course using two massive online open courses (MOOCs) and a flipped classroom model.
Here, we define flipped classroom in our context as a teaching format where the majority of the content is
transferred outside of scheduled class time via an interactive MOOC, and face to face time is used following this
content for consolidating knowledge and reflecting on learning during workshops and labs.
Teaching and learning about volcanoes is of public and professional interest, particularly in countries with
significant volcanic risk like Aotearoa New Zealand (NZ)[1]. MOOCs are a method where both public, professional,
and institutional audiences can be reached (Rodrigues-Silva & Alsina, 2024) and skills important to become
effective volcanologists can be taught. Flipped classrooms provide a scenario where content is delivered outside
the classroom at students' own time and pace, and that "homework" is turned into more active learning that takes
place in the classroom (Bergmann and Sams 2012), although the flipped model is variably applied and assessed
(Kapur et al., 2022). Where flipped classrooms are combined with interactive online material (Wang and Zhu,
2019, Forbes et al., 2023), students can be given opportunities for adaptive expertise and distributed practice using
workshops and laboratories where feedback and reflection are used to cement learning of skills. However, both
MOOCs and Flipped classrooms have challenges. MOOCs have low completion rates and frequently lack
meaningful peer and instructor interaction (Khalil, H. & Ebner, M. 2014, Kurtz et al 2023), in contrast flipped
classrooms frequently require very high levels of "buy in" from both instructors and students (Collopy & Arnold
2009). Studies where MOOCs and flipped classrooms are combined report some benefits over the stand-alone
models (Ghadiri et al., 2013).
The aims of the study are to (1) describe the development of a flipped classroom MOOC targeting bicultural
competence and science communication skills, (2) report students' reflections on learning in 2021, and 2022, by
coding reflections for comments relating to bicultural competence and science communication. (3) Discuss how
the students' reflections relate to specific course developments by comparing the findings from 2021 and 2022
student reflections and focus groups.
**2 Literature context.**
**2.1 Geoscience Skills.**
Volcanology courses exist as part of a geoscience program where course development, such as introducing new
skills, is achieved by mapping knowledge, skills and attitudes across courses to ensure graduate learning
objectives are met that service the geoscience workforce (e.g., Mosher and Keane 2021). The required skills from
geoscience employers are communicated to faculties and professional bodies via research on job advertisements
(Shafer et al., 2023), focus groups (Nyarko and Petcovic, 2022), and working groups with academics (Mosher and
Keane, 2021). These studies show the skills most valued by employers; specifically writing, field and data
collection, planning, communication, teamwork and interpersonal skills. Therefore, it is important to consider
which of these skills are currently being taught within the curriculum (Keane et al., 2022; Mosher et al., 2014;
Viskupic et al., 2021), so we can identify the skill deficits that need to be taught.
A general survey of the workforce highlighted geoscience skills within geoscience courses at undergraduate
geoscience programs (Viskupic et al., 2021). The survey reported that geoscience skills (e.g. rock description),
data skills, and communication skills were commonly practiced across many courses, although it should be noted
that the communication skills reported were around communicating with peers and the instructor, and not
specifically relating to communicating with the public or those outside of geosciences. Complementary to this, a
status of geoscience graduates report (Keane et al., 2022) highlighted three areas for improvement amongst our
geology graduate students: 1. working across cultures; and 2. Communicating with the public; and 3. Working in
interdisciplinary teams. This report, coupled with ongoing curriculum reform at the University of Canterbury,
provided motivation to develop, implement and research the integration of these skills within an existing 3rd year
volcanology course.
A range of practice-oriented, authentic, and/or work integrated tasks and assessments have been shown to be
effective at developing graduate attributes in education and nursing (Gulikers et al. 2004; Karunanayaka and
Naidu, 2021). These tasks range from work placements, fieldtrips, and simulations, to practice oriented or
authentic assessments (Kaider et al. 2017). Work placements and fieldtrips have been shown to be effective
authentic experiences aligning with desired skills (Miller and Konstantinou, 2022), but they are also time and
resource intensive, expensive, and not always equitable experiences (Kaider et al., 2017). Simulations and virtual
fieldtrips have been used which can be particularly affective when coupled with authentic assessment to augment,
achieve added value, or provide alternatives (Watson et al., 2023).

**2.2 Volcanology learning and teaching research.**

Recent research into teaching and learning in volcanology provides evidence for research gaps (Dohaney et al.,
20223a, 2023b). Volcanological learning and teaching research at a Tertiary level has focussed on research skills,
and field skills, followed by volcano monitoring, communication, teamwork and quantitative skills (Dohaney et
al., 2023a 2023b). In addition, a gap in research relating to addressing a lack of diversity is becoming increasingly
recognised and hence addressing this is a burgeoning area of research in geosciences (Gates et al., 2019; Mogk,
2021) and volcanology (Dohaney et al., 2023a; 2023b). Communication skills have been typically taught using
role play exercises and simulations (Harpp and Sweeney 2002; Nunn and Braud 2013; Barclay et al. 2011;
Teasedale et al. 2015; 2018; Dohaney et al., 2015; 2018) but also incorporated into lectures and labs (Whittecar,
2000; Gonzales and Semken, 2006).

**2.3 Cultural sensitivity in geoscience education.**

As much geoscience is landscape and place focused, it is inherently linked to culture (e.g. sense of place, Apple
et al., 2014) and cultural considerations can be crucial when working in the geosciences (Mosher and Keane,
2021). Despite this, working across cultures is rarely taught in the geosciences (Mosher and Keane, 2021).
Coincidentally, or consequently, diversity is low in geosciences compared to other sciences. Geoscience education
needs to be culturally responsive by explicitly centering Indigenous students, addressing racism, indigenous
identity, sovereignty, and data sovereignty (McKinley et al, 2023). Key strategies for indigenous student success
are multi-faceted, layered support, underpinned by the principles of respect, relationships, and responsibility
(Milne et al, 2016). Successful Earth science curricula for indigenous learners include outdoor education, a place
and problem-based structure, and the explicit inclusion of traditional indigenous knowledge. (Riggs, 2005).
Despite this, field trips are frequently cited as a barrier to indigenous students, due to family or tribal commitments
(recognising this will also impact other field-based disciplines) and/or general insensitivity to traditional
knowledge around place, (Marín-Spiotta et al., 2020; Carabajal & Atchison, 2020) and in particular around places
to be visited. Similarly, indigenous students face challenges on placement, including racism, discrimination,
misrecognition, and misrepresentation, and the importance of relationships for positive experiences (Pallas et al,
126 2022).

A recent study reports that culturally aware teachers, mentors or practitioners are an important factor in students
choosing the geosciences as a career (Todd et al., 2023). Appropriately incorporating traditional knowledge and

mentors into geoscience curriculum can improve communication and collaboration across disciplines and cultures, and encourage creativity and problem solving (Smythe et al., 2017; 2020). Indigenous research frameworks can enhance higher education by promoting relationality, multilogicality, and promoting equitable practices in research, teaching, mentoring, and organizational leadership for indigenous students. (Reano, 2020)

Recent research in volcanology education has emphasized the use of authentic voices to teach cultural sensitivity and indigenous knowledge across many cultures, particularly where volcanoes hold specific significance (Saha et al., 2021). Cultural competence is an area of educational focus in Aotearoa NZ, as workplaces are increasingly seeking employees with this skill set as the country strives to draw from all available knowledge, uphold treaty of Waitangi obligations and equitable educational outcomes for Māori and non-Māori. Māori, the indigenous people of Aotearoa NZ have their own knowledge system (Mātauranga Māori) and part of this are keen observational and generational understandings of their local area and the history of past volcanic activity (Cashman & Cronin, 2008; King et al., 2008; Tapuke et al., 2019). Regrettably, Mātauranga Māori has been either exploited or marginalized in science education (McKinley, 2005; Smith, 2012; Smith and Richie, 2013). Historically institutional racism has often attenuated Māori experiences in science education leading to under achievement in traditional measures of learning for Māori students and students from low socio-economic areas (Macfarlane & Macfarlane, 2018). Braiding of Mātauranga Māori with geology can thus lead to increased public preparedness and understanding of these natural processes (Bretton et al., 2018; Gabrielsen et al., 2018; King et al., 2008; Pardo et al., 2015; Swanson, 2008; Tapuke et al., 2019).

The teaching of volcanology frequently underutilises indigenous knowledge sources. The new courses described in this paper highlight the importance of weaving indigenous knowledge of areas studied, and the benefits that could come from shared and woven knowledge. This work builds on approaches and relationships outlined and defined in Saha et al. (2021, 2022). Indeed, the bicultural content used in the course in our research was in the form of a virtual fieldtrip, and several videos were reused and repurposed from such previous work, as well as additional videos recorded or sourced.

## 2.4 MOOCs and flipped classrooms.

The underlying concept of a flipped classroom is that the passive component of learning, the content delivery, is done before class, and the active component of learning, discussion, problem solving and collaboration is done with peers and instructors (Chen, 2016; Tan et al., 2017; Karagöl and Esen, 2019). "Content" (reading or videos) is delivered outside the classroom, and "homework" (problem solving with peers and instructors) is done in the classroom (Bergmann and Sams 2012). However, application and assessment of the flipped classroom model is highly variable, and a meta-analysis of data suggests that contrary to the underlying premise, it is not the active learning component of flipped classroom that drive measurable learning effects. Interestingly, the use of an additional lecture after online content shows a significant measurable learning effect (Kapur et al. 2022). Kapur et al. (2022) suggest a Fail (allow students to struggle with a problem), Flip (content delivered), Fix (misconceptions are explored) and Feed (feedback from students and instructors) model. Although this model is yet to be thoroughly tested, it emphasizes the role of allowing students to struggle, before or during content delivery when flipping the classroom.

On the other hand, MOOCs are a rapidly growing global phenomenon designed to make education globally accessible and allow students all over the world to learn from the world's best educators. There are many MOOC

platforms, and most courses consist of a format of short 3-8 min videos, and a series of questions and discussion boards, variably moderated by instructors and teaching assistants. However, MOOCs have high levels of students not completing courses and students who are disengaged with content. Small open online courses (SOOCs) also exist to address some criticisms of MOOCs, whereby small cohorts can more easily develop meaningful peer and instructor interactions.

Massive open online courses (MOOCs) and flipped classrooms can be seen as occupying two end members of the education spectrum in terms of individualised learning. MOOCs are designed to maximise the instructor reach, by making material accessible to a great number of students. Flipped classrooms were suggested as an alternative to the traditional classroom, as a methodology to promote active and tailored learning in classrooms by increasing instructor and student interaction. The learning experience in a MOOC invariably ends up being uniform and less personalised, whereas in a flipped classroom setting learning strives to be as individualised and personal as is practical. In our model, we drew from Māori education pedagogies to merge the advantages of the MOOC and flipped classroom formats. We deliver accessible online MOOC content with novel digital assessments and activities, in addition to face-to-face labs and flipped style workshops with the goal of developing lecturer-student-peer relationships and skill learning through reflection, discussion and connection to online environment. The benefits of working face-to-face and building lecturer-student-peer relationships are well established and highly effective Māori educational pedagogical techniques – kanohi-ki-te-kanohi and whanaungatanga respectively (Kana & Tamatea, 2012, Bishop et al. 2014)

The changes to the course discussed and presented here were developed and implemented before and during the context of COVID-19, and at a time when University of Canterbury had recently signed up to become part of global MOOC platform edX. The COVID-19 pandemic has shown a growth in online and blended learning, and with it a growth in the use of MOOCs (Aristovnik, et al. 2023). The post COVID-19 environment has seen a strong global demand for flexible blended courses, providing both flexibility of online content but also providing opportunities for face-to-face interactions when conditions allowed.

**3 Study setting and population.**

**3.1 Course information.**

The course that was the focus of the research is an elective undergraduate 3rd year Volcanology and Magmatic systems course, part of a geology BSc degree at a research university (Watson et al., 2022; 2023). The volcanology component of the course, the focus of this study, was redeveloped in 2022 following successful implementation of an Iceland virtual fieldtrip module (Watson et al 2022; 2023). The Iceland virtual fieldtrip formed the basis for MOOC development on the edX platform.

The redevelopment was driven by the instructor and informal conversations with Māori students in earlier iterations of the course, and the realisation that there was a missed opportunity to develop skills relating to science communication in the bicultural context of Aotearoa NZ. Of relevance is the strategic aims of University of Canterbury towards upholding and uplifting Te Tiriti o Waitangi (The Treaty of Waitangi). This includes the inclusion of Mātauranga Māori (Māori knowledge) and promoting the Bachelor of Science graduate profile of bicultural competence and confidence as essential skills in a multicultural Aotearoa NZ. The course instructor identified improving this outcome of the course as a key goal for this project.

The course development was made possible by a University of Canterbury program to foster professional
development, scholarship of learning, and leadership (University of Canterbury, 2024). Kennedy and Davidson
were provided with Distributed Leadership in Teaching Practice (DLTP) fellowships to explore the use of MOOCs
as tools to help flip the classroom at the University of Canterbury to provide flexible learning solutions for
students. The fellowships provided resources and time bought out from regular teaching research and admin duties
to develop a second MOOC, implement skills focussed assessment, and a model for the university to use MOOCs
to deliver online content and facilitate flipping of the classroom.
No student demographic data was directly collected for this research. However, we approximate gender
proportions from class enrolments and ethnicity from a yearly university-wide survey to provide readers with an
approximation of class demographics. In 2021, 48% of students identified as women, 50% of students identified
as men and 2 % as gender diverse. In 2022, 57% of enrolled students identified as women and 41% identified as
men and 2 % as gender diverse. In the university wide survey, of university students that were in $3^{rd}$ year geology
programs from 2019–2021, 73% of the students were of European descent, 16% of Māori ancestry, 3% were of
Asian descent, 3% of Pacific origin and 3% had other unspecified ancestry.
It is also worth mentioning the impact of the COVID-19 pandemic on these two study populations. Neither cohort
was directly impacted by COVID lockdowns during the implementation of this course, although it is worth noting
that the 2022 cohort had a larger proportion of their degree affected by COVID-19, particularly missing out on
several face-to-face fieldtrips in other related courses as a result of covid lockdowns during other semesters of
their study.
**3.2 The development of the course -Learning goals, implementation, and materials.**
The course was significantly changed by adopting a flipped classroom MOOC. We used the opportunity offered
by these changes to intentionally target bicultural competence and science communication skills. We used a
constructive alignment approach, a method where we used our course learning goals to link all assessments (online
content, laboratory exercises and workshop questions) ensuring that all learning is tied back to our original desired
outcomes for students taking the course. This was additionally motivated by University strategy to provide student
focussed, flexible, accessible education for all students. In 2021, the course learning goals were as follows:
1.   Realize the importance of igneous rocks in geology and to society.
2.   Identify and classify igneous rocks and their geological environments.
3.   Use geochemistry to explain why magma is generated, diversifies and erupts.
4.   Use geochemical data, thin sections, and maps to reconstruct the magmatic and volcanological histories.
5.   Discuss physical volcanological processes with relevance to magma properties.
6.   Describe volcanic rocks in the field using examples from Iceland and New Zealand.
In 2022 this new learning goal was added:
7.   Communicate science with different audiences and appreciate the value of Māori knowledge.

Baseline data was collected in August-October 2021. During this period, the course consisted of 6 weeks of volcanology content, two interactive 50 min lectures a week (using in class exercises and live multiple-choice quizzes), and a 2.5 hr hands-on laboratory and workbook (Table 1). The last two weeks of the course was devoted to the Iceland Virtual Fieldtrip, an interactive online module and two flipped classroom style workshops (Table 2). Students wrote a final summative exam during exam week. Course level learning goals focused around observing and explaining volcanic textures, landscapes, and processes, and interpreting eruption mechanisms and histories (Watson et al., 2022,2023). Specific learning goals were outlined in each lecture, laboratory, and online module.

The 2022 version of the course underwent a year of redevelopment working with online learning advisors, Mātauranga Māori advisors, and working with a community of practice of other DLTP fellows. Online learning advisors helped us design assessments and exercises that aligned to the learning goals and made use of functionality embedded in the edX online learning environment. The edX learning environment provided a range of assessment options with functionality that went beyond quiz questions. In addition to designing quiz type of assessments that provided instant feedback to students, we were able to incorporate peer assessment and reflection to promote engagement and learning at higher levels. The platform also enabled a seamless presentation of content in multiple ways such as text, video (including 360 video), audio, as well as interactive content for students to interact with in real time (e.g. interactive graphs, virtual simulations, interactive maps).

For cultural content and assessment design we worked with cultural and Mātauranga Māori advisors with whom we had strong existing relationships that had been carefully built and supported through research grants. Previous research showed that shared *relations* and *values* were crucial to create *space for sharing* where challenges and emerging understandings could be repositioned (Saha et al., 2022). Through discussion with our cultural and Mātauranga Māori advisors we obtained permission to reuse video segments mostly recorded for other purposes (e.g. Saha et al., 2022). In addition, we worked closely with the Faculty of Science kaiārahi, literally translated as the canoe steerer, but meaning (in this context) a cultural teaching and learning advisor. She helped us embed cultural content and design culturally appropriate assessments to go along with the videos provided by our cultural leaders and Mātauranga Māori advisors. We worked hard to embed the cultural aspects with assessment throughout the course to avoid tokenism, by valuing the content through assessment and reflection.

Similar to 2021, the redeveloped course in 2022 consisted of 6 weeks of teaching, however now students were expected to complete weekly 1.5-3 hrs worth of interactive online MOOC virtual fieldtrip work in their own time and attend both a 50 min flipped workshop and 2.5hrs of lab work (Table 1). In lieu of a final exam, students completed an applied science communication project which was handed in during exam week (Fig. 1). This change in assessment reflected the shift towards achieving the new skills-based learning goals.

The lab content and work were identical for both groups, and the learning goals were still focussed around observing, and explaining volcanic rocks and landscapes, and interpreting volcanic histories and mechanisms. As mentioned earlier, the key difference was the additional learning goal introduced into all the online modules and workshops, focussing on the skill of science communication to diverse audiences and around developing bicultural competence. Space to achieve these extra learning goals was made by reducing the number of international examples of volcanoes covered in lectures and by focussing on New Zealand and Iceland only.

Both the 2021 and 2022 versions of the course had online content that is interactive, with 360 videos, 3D rocks
and 3D landscapes, i.e., active, engaging online volcano science content. 360 videos and virtual rocks and
landscapes necessitated students to manipulate 3D space, and most activities have multiple choice, or drag and
drop questions with feedback provided for incorrect answers that guide students to think again or re-evaluate their
thinking in a particular direction (Table 3).
In 2021, the online content was only used during the last 2 weeks of the course. In 2022 the online content was
every week, and every module in the 2022 iteration of the course ended with an applied science communication
mapping activity. These skills orientated additional activities were introduced in the online content, in the
workshops and as an additional question in the laboratory workbooks. Some of the online science communication
tools in 2022 version also featured interactive online drawing exercises and peer assessment of other students'
answers, with marking rubrics that in most cases assessed cultural considerations (Table 3).
In 2022, we developed our flipped workshops (Table 2) to systematically incorporate exemplars of students'
online contributions, interactive questions used to promote mental ramp-up for students (Karpur et al, 2022) and
added focus on communication skills in the workbook questions and in class discussion. Additionally, in 2022, at
the end of each module students were asked to rate how confident they were to achieving learning goals and to
justify their responses at the end of each module (Fig. 2). This was implemented to guide the workshop part of
the course, where the instructor would review the student responses and focus on learning goals where students
were less confident (Fig. 2). Therefore, the workshop consisted of part lecture, which focussed on learning goals
where students were less confident and part reapplication and mastery of content in a different context through a
question that needed to be answered in a workbook.
In summary, the 2021 version of the course, had many elements of active learning in lectures, labs, and online
content but lacked the learning goal (7) Communicate science with different audiences and appreciate the value
of Māori knowledge. In 2021, only the last two weeks had a form of flipped classroom. In 2022, the class was
truly flipped aligning better with recent models of flipping of the classroom (e.g. Kapur et al., 2022), allowing
students to fail, and reflect.

## 4 Methods

The research used qualitative evaluation of students' responses to questions where students were asked to reflect
on their learning following a similar methodology to Engel et al., (2022). The study was reviewed and approved
by the university's human research ethics committee (Ref: 2021/116).

### 4.1 Data sources.

We used three different data sources to complete the qualitative evaluation. The first is a student reflection that
students completed towards the end of the course. The second data source are focus group interviews that were
completed after each course. The third data source are student artifacts from the online part of the course.
In both 2021 and 2022, at the beginning of the final laboratory session of the course, students were asked to
complete four reflection questions related to their learning in the course. All students were asked to complete this
questionnaire as part of their normal coursework. 21 students agreed to participate in the research (and thus share
their reflections) in 2021 and 27 students agreed in 2022. This research uses two of these questions as data sources:

Q1. What did you learn in this course and why is it important to you and/or your potential career? Q2. Has this course influenced your bicultural competence? (Table 4)

The timing of the questionnaire was immediately after the course content, although before most students had completed their projects. This offered the students a tangible and immediate opportunity on which to reflect on whether the course had achieved its intended learning goals. The reflective questionnaire served both as a means for students to consolidate their learning and as a data source for our research questions. Reflective questionnaires and journals are a common method in Science, technology, engineering, and mathematics (STEM) education research (Boyle et al., 2007; Scott et al., 2019; Treibergs et al., 2022). The first question offered an open-ended opportunity for students to think about what they learnt without being prompted towards thinking about learning goals or specific skills. The second question was targeted towards the specific learning objective of bicultural competence which aligns with a university wide graduate attribute.

Focus groups were held after the last week of class both years. The focus group interviewer asked several questions related to the course changes (Supplemental A). The main questions relevant to this research was:

- How has the course affected the way you feel/think about your bicultural competence and confidence?
- How has the course affected the way you feel/think about your science communication skills?

The focus groups were run after the course had ended but before the final exam or project was completed. 10 students participated in 3 focus groups in 2021 and 7 participated in 2 focus groups in 2022.

We also used student artifacts from to courses as a source of data. In 2022, at the end of each module students were asked to rate how confident they were to achieving learning goals and justify their responses. These responses, as well as other responses to open ended discussion questions throughout the course, were additionally analysed to explore whether specific part of the course led to perceived improvements in achieving the learning goals relating to communication skills or bicultural competence. 6 end of module questions and 11 discussion questions were analysed.

**4.2 Data Analysis method**

**4.2.1 Reflection questionnaire**

Students' reflection responses were coded by breaking down the two questions into sub-questions to help analysis (Table 4). Each questionnaire was then coded according to these sub questions using coding categories. For example, Question 1 of the questionnaire was simplified down three sub questions "What was learnt?", "Is what you learnt important to your future career?", and "Is what you learnt important to you personally?" (Table 4). Student responses to these questions underwent a first order coding content knowledge, skills or attitudes.

An example of how an answer is coded is shown in a response to Question 1 from the questionnaire below.

*"This section of the course has taught me heaps! I loved learning more about the different*
*types of volcanic eruptions, how they form, and the hazards associated with different*
*eruptions. I can see how understanding these fundamental concepts will be valuable going*
*forth into a geologist career. As well as learning about geology, this course also*
*strengthened my ability to be curious and excited about things and ask questions. It was*
*very eye-opening hearing Ben's reflection of the Whakaari disaster, as before hearing his*

*perspective I had never considered this implication between science research and human*
*safety of a tourist destination."*

In this quote, the student stated that they have learnt about content knowledge including different types of volcanic
eruptions, how they form, and the hazards associated with different eruptions. They also mention that the course
gave them an understanding of fundamental concepts that would be valuable for a geology career. Lastly, they
state that the course helped strengthen their curiosity and gave them excitement to ask questions. It also changed
their perspective on science research and human safety at a tourist destination such as Whakaari. This answer was
marked as a student having gained knowledge of a factoid and changing their attitudes towards learning and
thinking. This student did not mention anything about skills gained. This method was utilised for both questions
in the questionnaire across all years of this study.

**4.2.2 Focus Groups and Discussion boards**

Focus groups were recorded and then transcribed. The questions asked during the focus group were formulated to
supplement the questions asked in the reflective questionnaires. These questions and their equivalent in the
reflective questionnaires are presented in Table 5. The discussion board data was analysed to track if a comment
related to biculturalism or science communication. Both the focus group and discussion board data are used to
supplement the data from the reflective questionnaires to clarify and drill deeper into the meaning of the data.

**5 Results**

We report course reflections on learning in 2021, and 2022, relating to learning skills, and specifically code our
analysis for bicultural competence and science communication. We also present focus group discussions and
specific student reflections within the course that related to either bicultural confidence, science communication
or specific pedagogies, activities, or course elements.

**5.1 Overall learning**

The analysis of student reflections of Q1. *What did you learn in this course, and why is it important to you and/or*
*your potential career?"* showed that from 2021 to 2022 there was an increase of 13% of comments on learning
that related to skills when compared to learning content or general attitudes (Fig. 3, Table 4). When these skills
were categorized by types of skills, students in 2021 were more likely to mention skills relating to data, or other
skills such as microscope skills, whereas in 2022 students specifically mentioned bicultural competence and
communication, as well as flexible learning skills (Fig. 4).
In 2021 students were more likely to mention content knowledge and other skills relating to the laboratories, for
example,

*"I learnt different aspects of volcanology and magmas, this is crucial in understanding*
*volcanic environments and deposits as well as using microscopes to identify different*
*minerals in thin section and understand how this can relate to magmatic environments.*
*This could be applied to many careers outside of volcanology, the skills taught in this*
*course are essential for any geologist."*

The student mentions that they learnt about different magmas (content knowledge) and using microscopes (Skills).
They also address the second part of the question and mention that the skills learnt in the course are applicable to
many careers outside geology and are essential for any geologists. Other students' responses focused on content
knowledge and several students and did not mention any benefits to their future career, e.g.:
*"The effects of a volcano on the surrounding area in the form of ash and bombs etc.*
*different types of magmatic flows and what moves them such as gas content and if they're*
*mainly juvenile etc."*

## 5.2 The learning environment

Although the reflection question focused on *what* was learnt, many students mentioned *how* they learnt, and this
was coded as a skill in both years analysis. Particularly, the freedom to work at their own pace was commonly
mentioned in the reflection exercise:
*Helps me be able to go at my own pace, and not having to sit in one spot and watch a 50*
*minute straight lecture which is very boring and mentally draining.*
*The online modules were a very different way of learning than I was used to, and I think I*
*can take lessons of time management, persistence and quizzing from it. I think that my time*
*management started poor, but found that when I was able to push through the temptation*
*for distraction…I also liked how the online work quizzed me after introducing a topic, and*
*I think that this is conducive to my learning*
*The modules allowed me to work at my own pace and better understand the material as I*
*could go back and re-watch videos or re-attempt an answer if I got it wrong, which helped*
*me figure out what I needed to focus on more within the modules and my learning.*
*Allowing us to re-attempt the questions and self reflect/mark showed that I learn from*
*making mistakes and emerging myself within a topic more which was helpful.*
*I have enjoyed the small quiz questions directly underneath the content that introduces*
*what the questions will be about, it keeps the knowledge fresh for the questions.*
This flexible blended learning environment was seen as a positive development, especially the ability to work at
their own pace. Some students identified their own growth in time management skills. The frequent quiz questions
associated with content is also mentioned positively several times.

Overall, after the intervention students identified communication and bicultural competence skills, as well as
flexible learning more often in their reflections, with less mention of content knowledge and attitudes.

## 5.3 Specific aspects of the course linked to learning by students

Student specifically regularly mentioned exercises in a positive manner. Student appreciated the models and maps
that were part of the course as per some student reflections (this is consistent with previous studies Watson et al.,
(2023):

*The online lectures really helped with outcrop descriptions and 3D visualisation. the use*

*of models and maps in this course was AMAZING, and I really found they helped my*

*understanding of the larger scale geological processes which the course was trying to*

*teach us.*

Students mentioned the specific assessment exercises, e.g in the reflection at the end of a module, this student mentioned an open answer question that was asked in that module:

*It also was directly linked to what we were learning about like with the geothermal*

*resource email to the Maori land owner, rather then randomly being brought in every now*

*and then unrelated to what was goin on…I also like that there were Maori experts in their*

*fields who were directly teaching some of the concepts, that was great.*

In the focus groups, students mentioned that they appreciated the guided method used to teach skills through applied exercises:

*I think those sorts of yeah**[exercises]***, like none of my other courses have really touched on*

*that and having that like guided approach through it and like I think it's definitely a cool*

*skill that I have like obtained um, cos it's not just yeah, like rote learnt knowledge*

Practicing the procedure in different contexts and with different target audiences was seen as a beneficial, helping cement concepts, according to this quote from a focus group participant:

*"Um, maybe just adding to, I think having the multiple different tasks, like with concepts,*

*so you constantly had to think about the science side and the like, bicultural perspective*

*but in different formats …. a Facebook post which is, you know super, .. and then you had*

*one where it was like an email where you sort of had to be like, okay this is a different*

*format but the same thing and then talking to little kids, you're not going to use the same*

*words, same terminology, the same approaches to all of those things."*

**5.4 Communication skills**

In 2022 students typically mentioned communication skills which was coded as either general communication skills or part of bicultural competence in the case of specific bicultural communication.

*"The biggest thing I learnt in this course way how to communicate scientific ideas to a*

*non-scientific community in a way that helps them understand the ideas without creating*

*distress or make them feel that their culture is not heard and appreciated. This is a skill*

*that I will use within my future career when dealing with any people, both coworkers and*

*people within the public."*

*"This idea of respectful communication is something that would be important wherever I*

*go, and is something that I hope to be able to practice in the future."*

These are typical examples of student responses in 2022. The student mentions that they learnt how to communicate (Science communication). They also mention culture (Bicultural skills). They do not mention any

content knowledge, however they do acknowledge that the communication and cultural skills they gained will be
important in their future career. Some students did not link their skills learnt to their future career but did mention
content knowledge and the importance of bicultural communication.
*"I learned a lot about Nz and Iceland volcanoes and how they compare to each other. I*
*also learned about some mitigation strategies and how to categorise different types of*
*volcanic eruptions. I also learned a new way of learning online through these videos and*
*answering the questions as I go through. I learned about incorporating Maori knowledge*
*and the importance of Maori involvement with geothermal projects."*
Student's reflections at the end of module also mentioned the same sentiment. In the next three quotes, students
discuss the value of an exercise where students were asked to comment on the following fictional social media
post: "**I have heard volcanoes erupt after earthquakes and I know Lyttleton volcano has had an explosive**
**history, I also read in the news that there are some hot springs that have got hotter since the earthquake, I**
**am not sure if I could cope with lava on top of everything- does anyone know if the volcano will erupt.**
**Posted by John B**":
*"I thought this was a very relevant module that taught me skills that I will definitely use. I*
*quite often see misinformation or posts similar to John's, and I usually avoid them because*
*I don't know how to approach them. This module gave me the skills to do so."*
*"I found this module rather enjoyable, a lot of the time social media can provide a lot*
*misleading information that can generate unwanted fear in people or provide incorrect*
*information to people that can then be passed on. To be able to politely critique a member*
*of the general public and guide them towards more reliable scientific information."*
*"I enjoyed this module as I have seen posts on social media that were not well*
*communicated, so learning better ways to communicate was nice to see."*
These reflections show that the students felt the exercises were authentic and relevant to their learning journey.
**5.5 Bicultural Competence and Confidence**
The reflection question *"Has this course influenced your bicultural competence?"* (Fig. 5) showed that in 2021,
only 43% of students thought their confidence was influenced, in 2022 this number increased to 93%. When the
student answers were coded to explore their answers, it was apparent bicultural competence in 2021 was
interpreted as also knowing Icelandic experience with volcanoes by many students (Fig. 6), which isn't a surprise
given that most of the virtual fieldtrip in 2021 was set in Iceland and featured Icelandic locals and narratives. This
is illustrated by the quote from a student reflection response:
*"I think the Iceland trip perhaps enhanced my bicultural knowledge on how other*
*communities deal with volcanic hazards."*

A typical student response from 2022 was:

*"This course has definitely influenced my bicultural competence. I have gained a better*

*understanding of Maori and Icelandic cultures and the importance of being culturally*

*sensitive when communicating information."*

*"Absolutely, this is the course that has gone most into it. In a lot of other courses I feel as*

*though it is only really mentioned at the start maybe fore a mihi and then is forgotten*

*about has the course goes on, but here it was brough through out the whole course which*

*was cool. It also was directly linked to what we were learning about like with the*

*geothermal resource email to the Maori land owner, rather then randomly being brought*

*in every now and then unrelated to what was goin on…I also like that there were Maori*

*experts in their fields who were directly teaching some of the concepts, that was great."*


The focus group transcriptions and discussion board responses also revealed the value that the course contributed
to the student's bicultural confidence (Table 5). One example from a focus group discussion:

*"Yeah, I think the communication side of it was probably the most beneficial that I got out*

*of the course, um, especially yeah I suppose interacting with like manu whenua Māori and*

*um, and even just how to, I suppose adapt your communication to particular audiences"*


*"I feel like this part of the course has been very inclusive of what is the indigenous*

*approach to this, what is the cultural understanding, how can we incorporate the sort of*

*more indigenous aspects into how we approach science sort of like with the braided rivers*

*approach."*

In this quote, the student shows a new appreciation for adapting their messaging to different cultures.
The following quotes from focus groups interviews in 2022 show that students value bicultural confidence and
competence skills:

*"…especially in New Zealand, it's so important to incorporate that indigenous knowledge*

*when it comes to how we approach science"*


*"I feel like this part of the course has been very inclusive of what is the indigenous*

*approach to this, what is the cultural understanding, how can we incorporate the sort of*

*more indigenous aspects into how we approach science sort of like with the braided rivers*

*approach…"*


*I think it is a good reminder that a bicultural approach is necessary, especially within the*

*work place. I really liked how Ben used the karakea, as I felt it tied the course up nicely,*

*beginning to end.*


Students were very interested in the actual content and expressed that they would like to get a deeper
understanding of the subject, for example these discussion board quotes:
*I found the Māori volcanology legends fascinating and I would love to learn more about*
*how they view volcanoes and how we can use a mixture of both Western and Māori*
*knowledge to inform hazards and risks.*

*Learning about the volcano family, particularly in the context of Maori mythology is an*
*interesting idea that we often don't get to experience as science majors. Very cool!*

*I also enjoyed the input from Dan, on the ways we can implement maori/ native cultural*
*information, as you currently don't see a large amount of scientific literature with*
*consideration of these kind of inputs.*

## 6 Limitations

The research and the course assessment were intertwined, for example the instructor was also one of the
researchers, and part of the assessment (the reflections and discussion boards) were used as research artifacts.
However, marks for reflections and discussion boards were only for completion, and the student answers were
anonymized before the instructor saw them (e.g. Watson, 2022) as per ethics agreement. Similarly, when the
research was presented to the students and their participation in the research was requested, the instructor was out
of the room as per ethics agreement. However, considering that the research relies on the students' perception of
their learning, it is possible that students' perception of what they were learning was influenced by the research.
Given that this study is a comparison between two years, and both years the research methodology was identical,
comparisons between both cohorts should be uninfluenced by the research.

## 7 Discussion

By comparing the results from the two separate classes, we can get some insight into the effect of the course
changes. The analysis shows that students in the post-intervention group were more likely to mention skills in
their reflections on what they learnt (Fig. 3). When analysed further, the skills that were mentioned were most
likely to be relating to communication, online environment, and bicultural competence. This increase in
mentioning skills aligns with the instructors' goals for the changes implemented in the course, which were
specifically to improve the communication skills and bicultural confidence of the students, (Fig. 4).
Overall, the student reflections show that the change from a lecture-based classroom setting to a flipped classroom
with an interactive, engaging, and pedagogically grounded online environment was an effective classroom
intervention. The 2022 iteration contained more interactive elements and functionality aligned to communication
and bicultural competence. It provided more authentic assessment and opportunities for deliberate practice (e.g.
Benjamin & Tullis, 2010), which are pedagogical approaches that are linked to improved learning. The delivery
of both 2021 and 2022 content was during the COVID pandemic although neither were affected directly by
lockdowns, the reflection questions analysed here did not address the impact of COVID pandemic on learning,
although this context is important to consider as has been shown to influence students and instructors' opinions
of online learning (Chakraborty et al. 2021).
The intervention contained additional exercises to encourage students to engage with class material outside of the
classroom and apply what they learned to real world situations that they expect to experience in a future career.
These exercises can be defined as authentic assessments (Kaider et al. 2024). Students linked these authentic
assessments with their perceived learning in the discussion boards, reflective posts, or focus groups where students
reflected on specific exercises linking these exercises to learning specific career useful skills. Students quotes
showed that they felt that the course provided them with opportunities to practice skills to communicate
effectively. They felt that these authentic assessment exercises could help them develop skills that could be useful
in future careers. Some quotes reveal that these skills were something that the students had already encountered
in their personal lives and therefore valued as authentic. These skills are not only related to the courses' learning
goals, and also can be linked to University of Canterbury's BSc graduate attributes of "Biculturally competent
and confident" and "Employable" specifically around "Communication". Although not directly related to a
specific geology career, they are, in essence, skills required to become an informed BSc graduate and citizen.
A clear change between both cohorts is the bicultural confidence context that students mentioned in their
reflections. In the 2021 cohort, most of the group discussed Icelandic culture when asked about bicultural
confidence and competence, however almost all students took it to mean Māori culture in the 2022. Although this
in not the case overseas (e.g Clark ,2006), in New Zealand, biculturalism specifically refers to the existence of
two distinct cultures, Māori culture and New Zealand culture, based primarily on values from British settlers
(Eketone, & Walker, 2015). This latter definition of biculturalism is relevant here and when interpreting students'
responses regarding bicultural confidence and competence in the reflective questionnaire in both years. That
students mentioned Māori culture less in 2021 is likely related to the lack of Māori experts and assessments
relating to bicultural confidence featured the 2021 version of the course, and therefore the students might have
felt that bicultural confidence in the context of the course did not specifically relate to Māori culture. The 2022
data shows that bicultural understanding was at least partially shifted and was related to the Māori experts featured
in the course and the related assessments. Improved cultural competency has been reported to enhances people's
well-being by bringing together indigenous and nonindigenous knowledge and practices (Eketone & Walker,

595    2015).

Students quotes overall showed a genuine appreciation for the Mātauranga Māori and bicultural content.
Reflections showed that the content in the 2022 version of the course felt authentic and better integrated in the
course compared to other courses. Students appreciated that the instructor lead by example by adhering to Māori
tikanga (customary practices) while delivering the course. Our model of MOOC, flipped classroom, and focus on
developing lecturer-student and peer relationships is an expression of Māori tikanga, and enabled students to
experience it through undertaking the course. For example, students experienced whanaungatanga (meaning
"creating cohort connection through relationship building" in this context) through the intentional relationship
building, and further by writing and sharing their pepeha, reading other students' pepeha, an activity that the
students highlighted in their reflections. Students also commented that they appreciated videos shared by our
cultural experts, where cultural values were frequently expressed such as kaitiakitanga (intergenerational
sustainable guardianship of the land) around the geothermal industry.

## 8 Conclusions

Our research describes the pedagogy behind our course and the critical roles that all the members of the team had
in course development. We then present and discuss data on students' perceptions of their learning and how this
relates to elements of the course.
Learning advisors guided us to produce engaging interactive activities on the edX online platform, these were
critical in allowing creation of activities that enabled deliberate practice of skills in a variety of assessment types.
Similarly, our cultural advisors who also delivered authentic content, providing essential mana and expertise in
cultural knowledge and how to design assessments that reflect and test this knowledge. These roles were essential
to achieve the learning associated with skills-based learning objectives in 2022, and this was in addition to the
critical roles of the instructor and 3D visualisation tools developer as discussed in Watson et al. (2022;2023)
Students in 2022 were more likely to mention communication skills, bicultural skills, or skills relating to flexible
learning when asked to reflect on their learning. Several students in 2022 specifically mentioned the newly
introduced authentic assessments and linking this to their skill learning. Some students also mentioned the
opportunity to practice skills in a variety of contexts and tools.
The team-based development of the flexible course, with multiple experts and repurposing of videos should
provide a template for the development of other courses with skills-based course learning goals. In addition, the
research supports the use of multiple flexible modes of authentic assessment to promote skills-based learning.
In summary, students' reflections showed that during the course they gained bicultural confidence and
communication skills. Our consideration of Māori tikanga, Mātauranga (knowledge) and values such as
kaitiakitanga (guardianship) and whanaungatanga (relationships) alongside scientific methods fostered the ability
to communicate science with a range of people with different academic and cultural backgrounds, which is
important in most careers in Aotearoa NZ and globally. We encourage other academics to uphold local indigenous
cultural perspectives when developing and delivering science courses.

**Foot notes**

[1]Although Aotearoa is a Māori name for New Zealand's North Island, to reflect the nations bicultural foundation
it is commonly and increasingly used in this way, e.g., Aotearoa New Zealand, or simply Aotearoa NZ, to mean
New Zealand.

**Author contribution:**

BK and JD designed the experiments and JD and KE carried them out. All Authors contributed to the design and
re-development of the course. BK prepared the manuscript with contributions from all co-authors.

**Competing interests:**

The authors declare that they have no conflict of interest.

**Ethical statement**

The study was reviewed and approved by the University of Canterbury's human research ethics committee (Ref: 2021/116).

**Acknowledgements**

We would like to acknowledge the contributions of Abby Susko as Kaiarahi Māori for the faculty of science in helping design of bicultural assessments and general tikanga. We would also like to acknowledge Kelvin Tapuke and Bubs Smith for their cultural guidance throughout the project. We also would like to thank Rob Stowell for his tireless help with video recording and scripting. We would also like to thank Mathew Stiller-Reeve, Siri Veland and one anonymous reviewer for providing invaluable feedback on this manuscript. Funding was provided for Kennedy through Ministry of Business Innovation and Employment, Endeavour project Beneath the Waves.

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

**Tables**

| Course run | Implementation and assessment | Research data |
|---|---|---|
| 2021 | 10hrs lectures with assessed in class exercises<br>15hrs laboratory work assessed workbook<br>4hrs online VTF and assessed online discussion boards<br>2 hrs Workshop in last 2 weeks of course:<br>Study for exam and 2.5 hr final exam | End of course reflection<br>Focus groups |
| 2022 | 12hrs online MOOC VTF online exercises<br>15hrs laboratory work and assessed workbook<br>5hrs flipped workshops and assessed workbook:<br>Science communication project | End of course reflection<br>Weekly reflections<br>Focus groups |

**Table 1: Course structure and research data from both runs of the course**

| 2021 | Informal discussion with group (10 min)<br>Handout with applied interpretative sketches and reflections. (35 min) |
|---|---|
| 2022 | Highlighting exemplar online responses from students(5min)<br>Reflection on confidence of achieving learning goals as guidance for workshop (5min)<br>Warm up question as mental ramp up(5min)<br>Mini interactive lecture based on content with low confidence (10min)<br>Two group workbook questions, designed to explore content and develop communication skills (20min) |

**Table 2: Workshop structure comparison from both runs of the course**


| Online content | Video | Multichoice | Discussion boards | Interactive | End of module |
|---|---|---|---|---|---|
| VFT (2021, last two weeks) | Instructor, and Iceland experts | Content focussed | Prior knowledge,<br>Reflection,<br>Sketching skills | 3D rocks,<br>360 video,<br>3D landscapes | Applied question (+one module with reflection) |
| MOOC (2022, every week) | Instructor,<br>Indigenous leaders and scientists<br>Iceland and NZ experts | Content and skill focussed | Prior knowledge<br>Reflection<br>Sketching skills<br>Communication skills | 3D rocks,<br>360 video,<br>3D landscapes<br>Mapping with communication and cultural elements | Reflection after specific goals achievement exercise. |

**Table 3: Details of online content.**

| Reflection Questions | Question aspects | Coding categories | Results | |
|---|---|---|---|---|
| | | | 2021 (Total n = 21) | 2022 (Total n = 27) |
| Q1. What did you learn in this course and why is it important to you and/or your potential career? | What was learnt? | Content knowledge (Factoids) | 21 | 23 |
| | | Skills, 2021 | 18 | 26 |
| | | Attitudes | 11 | 4 |
| | Is what you learnt important to your future study/career/personally? | Yes | 17 | 21 |
| | | No | 1 | 0 |
| | | Not Stated | 3 | 4 |
| Q2. Has this course influenced your bicultural competence (BCC)? | Did the course improve your BCC? | Yes | 9 | 25 |
| | | No | 8 | 1 |
| | | Unsure | 3 | 1 |
| | | Not stated | 1 | 0 |
| | What kind of cultural knowledge was improved? | Māori | 5 | 17 |
| | | Icelandic | 8 | 1 |
| | | Other | 1 | 7 |
| | | Not stated | 12 | 2 |

**Table 4. Coding methodogy. The numbers reported in the columns are the number of students that mentioned the**
**category in their reflections. n is the total number of answer.**

| Focus Group Questions | Reflective Questionnaire Equivalent |
|---|---|
| How has the course affected/influenced/helped/assisted your learning in volcanology/geology? | **Q1.** What was learnt? (Facts/Attitudes) |
| How has the course affected the way you feel/think about your science communication skills? | **Q1.** What was learnt? (Skills) |
| How might your experience with the course help you in the future? | **Q1.** Is what you learnt important to your future study/career/personally? |
| How has the course affected the way you feel/think about your bicultural competence and confidence? | **Q2.** Has this course influenced your |

Page 27

| | bicultural competence (BCC)? |
|---|---|
| | |

**Table 5. Focus group questions mapped onto the reflective questionnaire questions**


**Figures**

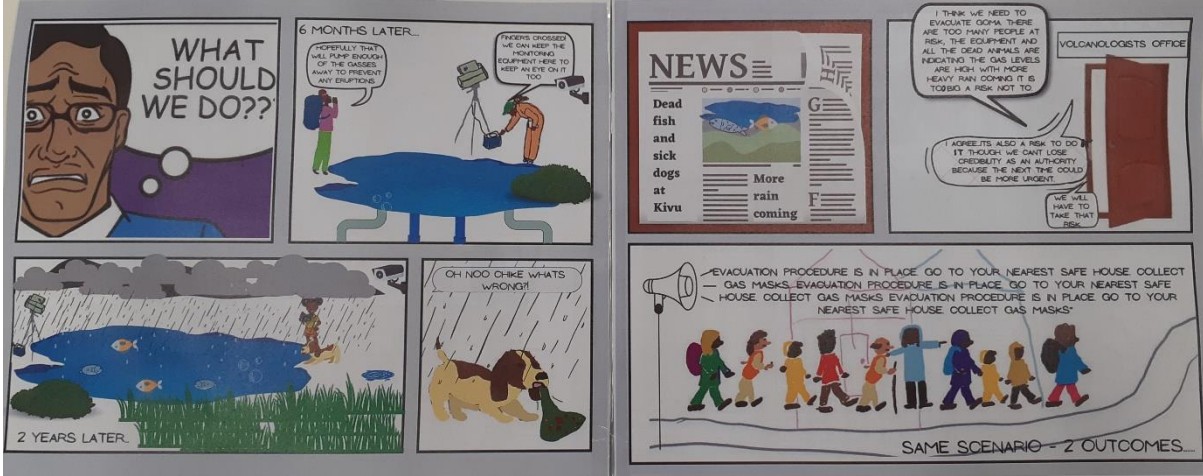


Figure 1. An example of two pages of a student communication project aimed at educating children about hazards
from gases in the city of Goma in the Democratic Republic of Congo.

|  | Very confident | Fairly confident | Not so confident |
| --- | --- | --- | --- |
| Locate Krafla in the context of the volcanism and tectonism of Iceland. | 65% | 31% | 4% |
| Record systematic and detailed observations of an outcrop and a rock of a typical obsidian subglacial eruption. | 46% | 46% | 8% |
| Record and compare systematic and detailed observations of the map-scale geology and geomorphology of tuff cones, lava domes and/or caldera volcanoes. | 42% | 54% | 4% |
| Use rocks to communicate your understanding of volcanic processes and products to a diverse group of primary school | 50% | 38% | 12% |

- I personally had real trouble looking at the outcrops as I feel like the quality wasn't ideal. I knew it was a dyke and it's structure, but it was difficult to make all the observations correctly even when I was looking really hard and trying my best. I really enjoyed the hazard exercise, as explaining the rocks to children really cemented that I knew the difference and could explain it in a simple way. I also didn't know there was such a thing called a rootless volcano, so that was super interesting to learn! I enjoyed having the exercise to identify them too as it made it a bit clearer what they looked like.



Figure 2. Example slide from the start of a workshop session in 2022, showing learning goals, a summary of
student self -reported confidence on learning goals for the module, and one example justification from students.
This slide shows how learning goals are constructively aligned by giving students the opportunity to self-report
and reflect on their achievement of the goals.


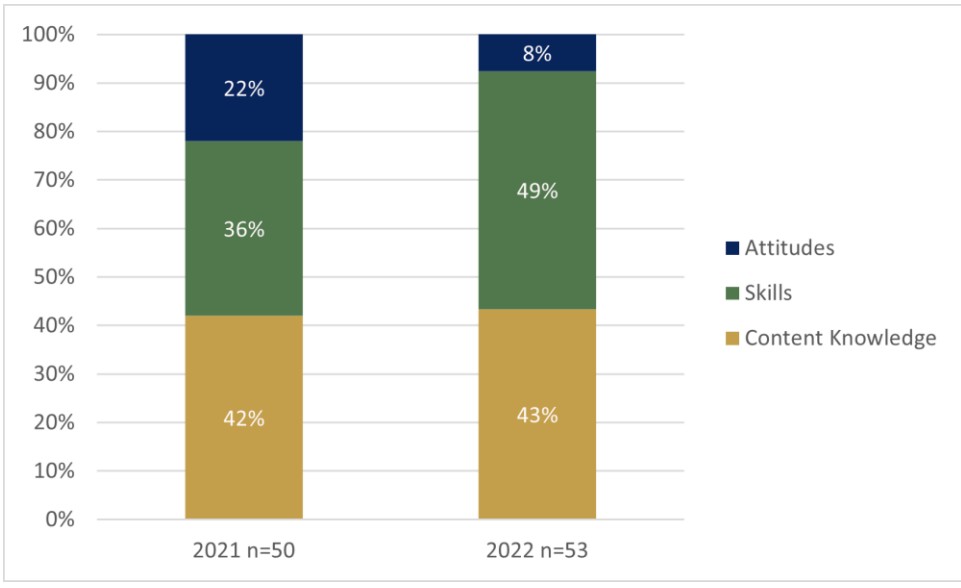


**Figure 3. Relative coding results from reflection data. n is total number of the code category mentions (one answer**
**might have multiple mentions).**

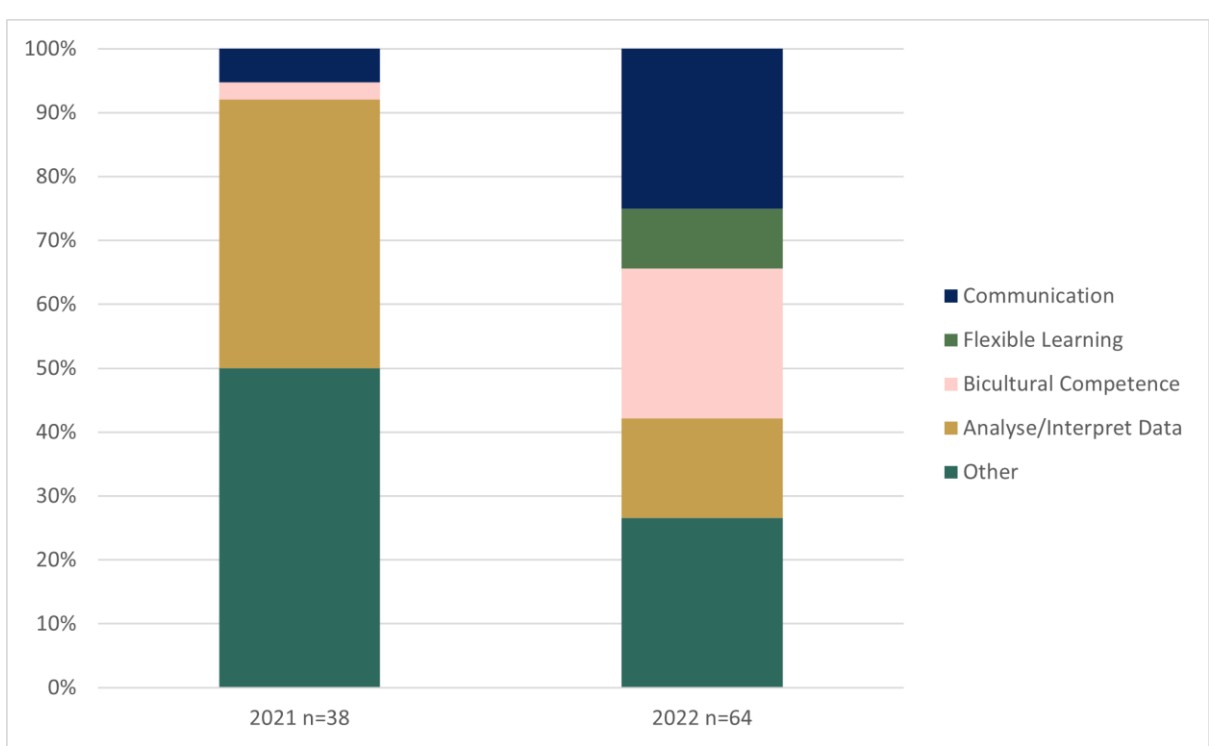

**Figure 4. 2021 and 2022 data comparison of skills mentioned by students in their reflections. (one answer might have**
**multiple mentions).**



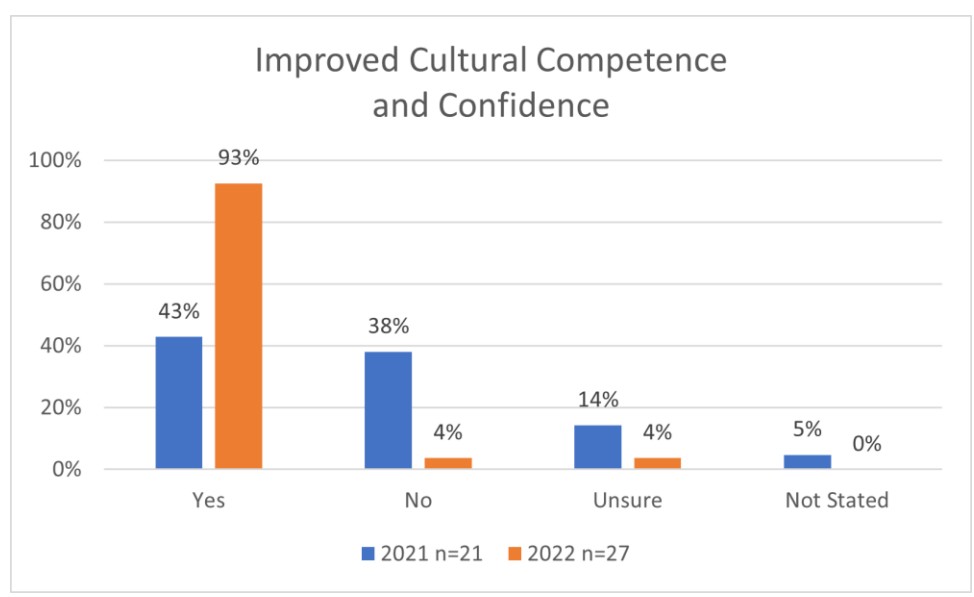

**Figure 5. Summary of cultural competence perception in student reflection questions**


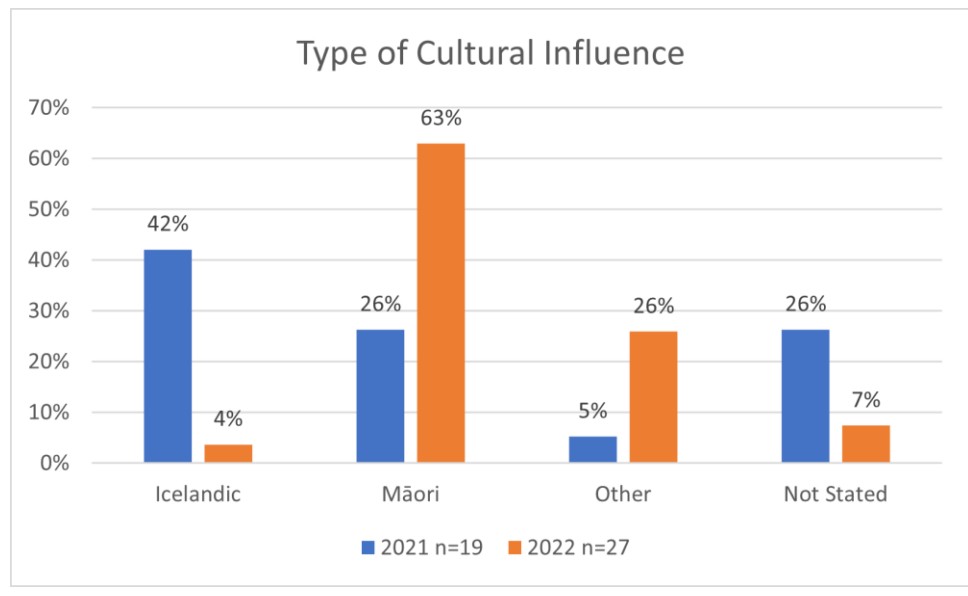

**Figure 6. Summary of cultural influence type mentioned in student reflection questions.**