# Peer review of "Incorporating science communication and bicultural"

_EGUsphere, 2024_

## Author Response (AR1)

Thank you for the thorough review of our manuscript. We provide here a detailed overview of the changes we have made to the manuscript following the review. Note that lines in this document refer to the pdf with the track changes turned on.

1. *Revise the abstract to avoid repetition and redundancy (both reviewers).*:

   We edited the abstract to remove unnecessary parts.

2. *Clearly define the term "flipping" earlier in the paper. This is a crucial concept for the study, and not all readers will be familiar with it (both reviewers).*

   We now define how we use the term"flipping" earlier in the text **(L.51-53)**: "Here, we define flipped classroom in our context as a teaching format where the majority of the content is transferred outside of scheduled class time via an interactive MOOC, and face to face time is used following this content for consolidating knowledge and reflecting on learning during workshops and labs."

3. *Provide more detail, or clarify the format, of the online workshops (as noted by reviewer 1).*:

   As well as additional context on how the class was run **(L.180-186 see additional comments from editor point 5)**, we also added some more specific context, a new figure **(Fig.2)** and a new table when describing how the workshops were run, including a new table **(L.296-298 & Table 2)**:

   "In 2022, we developed our flipped workshops to systematically incorporate exemplars of students' online contributions, interactive questions used to promote mental ramp-up for students (Karpur et al, 2022) and an added focus on communication skills in the workbook questions and in class discussion."

4. *Explain how the online format influenced the transmission of Mātauranga Māori (reviewer 1). This also ties in with reviewer 2's comment on how "relationships, values, and sharing are cultivated within the course while training students in technical skills.".*

   We added two sections explaining more about how Mātauranga Māori values were embedded in the online course structure.

   *(L.180-186):*"In our model, we drew from Māori education pedagogies to merge the advantages of the MOOC and flipped classroom formats. We deliver accessible online MOOC content with novel digital assessments and activities, in addition to face-to-face labs and flipped style workshops with the goal of developing lecturer-student-peer relationships and skill learning through reflection, discussion and connection to online environment. The benefits of working face-to-face and building lecturer-student-peer relationships are well established and highly effective Māori

educational pedagogical techniques – kanohi-ki-te-kanohi and whanaungatanga respectively (Kana & Tamatea, 2012, Bishop et al. 2014)"

*(L.607-614)* "Our model of MOOC, flipped classroom and focus on developing lecturer-student and peer relationships is an expression of Māori tikanga, and enabled students to experience it through undertaking the course. For example, students experienced whanaungatanga (meaning "creating cohort connection through relationship building" in this context) through the intentional relationship building, and further by writing and sharing their pepeha, reading other students' pepeha, an activity that the students highlighted in their reflections. Students also commented that they appreciated  videos shared by our cultural experts, where cultural values were frequently expressed such as kaitiakitanga (intergenerational sustainable guardianship of the land) around the geothermal industry."

Additional comments from the Editor:

5. *Line 174: "Massive open online courses (MOOCs) and flipped classrooms can be seen as occupying two end members of the education spectrum." While this is an interesting observation, it would be helpful to clarify how these two methods were concretely mixed in this study. A clearer description would enhance the reproducibility of this work in other contexts.*:

   We provide additional general details on the course format run **(L.180-186 see point 4 above)**.

6. *Line 223: The term "constructive alignment approach" is mentioned but not revisited in the paper. Consider removing it unless it is discussed further*: We discuss out approach in more detail **(L.229-231)**:

   "[…] a method where we used our course learning goals to link all assessments (online content, laboratory exercises and workshop questions) ensuring that all learning is tied back to our original desired outcomes for students taking the course"

7. *Line 245: If Gagné's 9 Events of Learning are used in your analysis or methods, make this explicit. If they are only mentioned in passing without further application or discussion, consider removing them.*:

   We chose to remove the reference the Gagne's events of learning, as suggested.

8. *Finally, reviewer 2 highlights an important point: the paper currently lacks a clear, inspiring message for an international audience. What can readers from other societies take away from this study to inform their own educational programs?*:

   We added a message to the conclusion of the paper **(L.633-638)**:

"In summary, students' reflections showed that during the course they gained bicultural confidence and communication skills. Our consideration of Māori tikanga (customary practices), Mātauranga (knowledge) and values such as kaitiakitanga (guardianship) alongside scientific methods fostered the ability to communicate science with a range of people with different academic and cultural backgrounds, which is important in most careers in Aotearoa NZ and globally. We encourage other academics to uphold local indigenous cultural perspectives when developing and delivering science courses."

Additional edits:

9. We included a discussion on how the pandemic affected the study, as suggested by reviewer 2 **L(573-577)**:
   "The delivery of both 2021 and 2022 content was during the COVID pandemic although neither were affected directly by lockdowns, the reflection questions analysed here did not address the impact of COVID pandemic on learning, although this context is important to consider as has been shown to influence students and instructors opinions of online learning (Chakraborty et al. 2021)."

10. We added a new figure as an illustration of the types of project produced by students in response to a reviewer 2 query.(**Fig 1)**

11. We have addressed various syntax and grammatical errors throughout the text, the changes have been tracked in the document attached.